# Fetal Growth Restriction: Comparison of Biometric Parameters

**DOI:** 10.3390/jpm12071125

**Published:** 2022-07-11

**Authors:** Carolin Marchand, Jeanette Köppe, Helen Ann Köster, Kathrin Oelmeier, Ralf Schmitz, Johannes Steinhard, Arrigo Fruscalzo, Karol Kubiak

**Affiliations:** 1Department of Gynecology and Obstetrics, St. Franziskus Hospital Muenster, 48145 Muenster, Germany; dr.karolkubiak@gmail.com; 2Institute of Biostatistics and Clinical Research, University of Muenster, 48149 Muenster, Germany; jeanette.koeppe@ukmuenster.de; 3Practice of Gynecology and Obstetrics, Schloßstraße 107-8, 12163 Berlin, Germany; hkoester@hotmail.de; 4Department of Gynecology and Obstetrics, University Hospital of Muenster, 48149 Muenster, Germany; kathrin.oelmeier@ukmuenster.de (K.O.); ralf.schmitz@ukmuenster.de (R.S.); 5Department of Fetal Cardiology, Heart and Diabetes Center North Rhine-Westphalia, 32545 Bad Oeynhausen, Germany; johannessteinhard@icloud.com; 6Department of Gynecology and Obstetrics, HFR Fribourg, Chemin des Pensionnats 2-6, 1708 Fribourg, Switzerland; arrigo.fruscalzo@h-fr.ch

**Keywords:** small for gestational age, fetal growth retardation, fetal biometry, fetal ultrasound, fetal sonographic parameters, fetal sonographic ratios

## Abstract

The aim of this study was to identify growth-restricted fetuses using biometric parameters and to assess the validity and clinical value of individual ultrasound parameters and ratios, such as transcerebellar diameter/abdominal circumference (TCD/AC), head circumference/abdominal circumference (HC/AC), and femur length/abdominal circumference (FL/AC). In a retrospective single-center cross-sectional study, the biometric data of 9292 pregnancies between the 15th and 42nd weeks of gestation were acquired. Statistical analysis included descriptive data, quantile regression estimating the 10th and 90th percentiles, and multivariable analysis. We obtained clinically noticeable results in predicting small-for-gestational-age (SGA) and fetal growth restriction (FGR) fetuses at advanced weeks of gestation using the AC with a Youden index of 0.81 and 0.96, respectively. The other individual parameters and quotients were less suited to identifying cases of SGA and FGR. The multivariable analysis demonstrated the best results for identifying SGA and FGR fetuses with an area under the curve of 0.95 and 0.96, respectively. The individual ultrasound parameters were better suited to identifying SGA and FGR than the ratios. Amongst these, the AC was the most promising individual parameter, especially at advanced weeks of gestation. However, the highest accuracy was achieved with a multivariable model.

## 1. Introduction

Abnormal fetal growth occurs in about 10% of pregnancies and is associated with increased perinatal morbidity and mortality [1]. It is an important predictor of pregnancy outcome and reflects the interaction between physiological and pathological factors affecting the fetus. Identification of growth retardation is an essential part of prenatal care because it is associated with higher risk of perinatal asphyxia, stillbirth, neonatal hypoglycemia, and other metabolic complications compared to appropriate-for-gestational-age (AGA) fetuses [2]. Previous studies showed that cerebral palsy is 4–6 times more common in neonates below the 10th percentile than AGAs [3]. Prenatal detection of fetal growth restriction (FGR) is an important factor in stillbirth prevention strategies, as up to 30% of cases are associated with FGR [4]. However, detection rates for FGR remain low, and the risk of stillbirth is increased eightfold [5]. In turn, identification of fetal growth restriction leads to a four-fold reduction in neonatal complications and deaths [6]. However, growth retardation also has long-term effects, including delayed neurological development in childhood and increased risk of cardiovascular disease, dyslipidemia, and diabetes mellitus in adulthood [7]. Small for gestational age (SGA) is defined as continuous fetal growth below a predefined percentile, usually the 10th. Not all SGA fetuses are growth-restricted; most are constitutionally small [8].

In contrast, fetal growth restriction (FGR) refers to a fetus unable to achieve its genetically determined growth potential due to fetal, placental, or maternal factors [8]. The difficulty lies in distinguishing FGR from SGA fetuses. Accurate prenatal identification of FGR enables appropriate prenatal monitoring, timing of delivery, and early neonatal treatment, reducing the aforementioned risks [9].

The most frequently used biometric parameters for assessing fetal weight include the biparietal diameter (BPD), the head circumference (HC), the abdominal circumference (AC), and the femur length (FL). The cerebellum is relatively resistant to chronic hypoxia due to blood flow centralization (brain sparing effect), resulting in the maintenance of blood supply to the brain at the expense of systemic supply. Furthermore, growth retardation of the fetus has minimal effect on size of the cerebellum [10,11]. AC reflects size of the liver and the volume of the subcutaneous plane. Both are affected by fetal malnutrition [12]. In addition, several authors have suggested that the ratio of transcerebellar diameter/AC (TCD/AC) is a suitable method for assessing fetal growth, especially in FGR [10,13,14,15,16,17]. HC/AC is another ratio and compares the preserved organ in the malnourished fetus to its most vulnerable organ and is of considerable value in identifying asymmetrical growth restricted fetuses [18].

As previous studies never compared all sonographic parameters and important ratios in one paper, the present study aims to compare various biometric parameters for identifying growth retardation and test the usefulness of standard ultrasound ratios, such as FL/AC, TCD/AC, HC/AC. The second aim was to create a regression model for predicting SGA and FGR to improve the current prenatal detection rate.

## 2. Materials and Methods

### 2.1. Study Design

The study was conducted retrospectively as a single-center cross-sectional study. We extracted the delivery details of 26,724 newborns with prenatal ultrasound examinations between 2001 and 2019 from the hospital database (Viewpoint ®, General Electric, Wessling, Germany). The study population was recruited with the following inclusion criteria: (1) singleton live pregnancies; (2) examination between 14 + 0 and 41 + 6 weeks with expected dates of birth determined by ultrasound using the crown-rump-length; (3) one fetal ultrasound per pregnancy. Estimated fetal weight (EFW) was calculated according to the formula IV proposed by Hadlock et al. [19].

Fetal size classification was based on EFW that was small (<10th percentile), adequate (10–90th percentile), or large (>90th percentile) for gestational age [20]. Following the German guideline, we defined our patient population as follows [8]: We categorized all fetuses and neonates with an EFW and birth weight below the 10th percentile as SGA, and all fetuses with an EFW and birth weight above the 90th percentile as LGA. FGR fetuses were defined by an EFW and birth weight below the 3rd percentile [21]. We used unisex percentile charts for the EFW and sex-specific percentile charts for the birth weight, as described by Hadlock et al. and Voigt et al. [22,23].

The exclusion criteria included multiple gestation, congenital fetal anomalies, and sonographic evidence of malformations, such as gastroschisis, skeletal dysplasia, or hydrops fetalis. We also excluded fetuses with differing categorisation of EFW and birth weight. As the focuses of this study were SGA and FGR, we excluded LGA fetuses (N = 77) from further analysis. For a more detailed analysis, we divided the fetuses into groups (AGA/SGA/FGR) and split the growth restricted fetuses into subgroups according to gestational age at examination date. We considered the gestational age to the exact day.

In order to avoid bias, only one assessment per fetus was included in the statistical analysis. The parameters TCD, BPD, OFD (occipitofrontal diameter), AC, and FL were measured for each fetus. The measurements were carried out in standardized procedures using standard ultrasound planes described in previous publications by Hadlock, summarized in a special report in the *American Journal of Obstetrics and Gynecology* [24]. HC was calculated as defined by Sniders and Nicolaides [25]. The TCD/AC, HC/AC, and FL/AC ratios were calculated by dividing the TCD, HC, and FL by AC and multiplying by 100 [12,26,27].

### 2.2. Statistical Analysis

Statistical analysis was performed using IBM SPSS Statistics 22 for Windows (IBM Corp. Somers. NY, USA) and SAS software V9.4 (SAS Institute Inc., Cary, NC, USA). Descriptive statistics were used to characterize the entire cohort and the individual groups (AGA/SGA/FGR). The inferential statistics are intended to be fully exploratory (hypothesis-generating) and are interpreted accordingly. Therefore, no adjustment was made for multiple testing. The *p*-values are regarded as noticeable if *p* ≤ 0.05. The association between SGA and FGR and ultrasound parameter/ratios was assessed with a logistic regression model. The respective ultrasound parameter/ratio, gestational age, and interaction term between parameter and gestational age were included in the models. A quantile regression analysis was performed to define the cutoff values, defined as the age-adjusted 10th percentile for ultrasound parameters (HC, BPD, AC, FL, TCD) and age-adjusted 90th percentile for ultrasound ratios (FL/AC, TCD/AC, HC/AC). All fetuses with the respective ultrasound parameter below the age-adjusted 10th percentile or ultrasound ratio above the 90th percentile were classified as SGA. FL/AC and HC/AC were standardized so that the magnitude of all ratios was approximately the same. The sensitivity, specificity, positive predictive value (PPV), negative predictive value (NPV), and the Youden index were calculated for this classification. For FGR, all analyses were performed analogously, including only fetuses with an EFW and birth weight below the 3rd percentile. A multivariable logistic regression analysis was performed for SGA and FGR including maternal age, weight, gestational age, and sonographic parameters (HC, BPD, AC, FL, TCD).

## 3. Results

Of the 26,724 data sets, 9292 examinations were included in the final analysis. Our criteria resulted in 8231 (89%) fetuses being classified as AGA, 1061 (11%) as SGA, and 255 as FGR (3%). The characteristics of the study population are displayed in Table 1, including the distribution of gestational age. In daily practice, screening of second-trimester fetuses is performed between 20 and 24 weeks. Accordingly, 53% of the patients were examined between 19 + 0 and 23 + 6 weeks of gestation.

Our analyses demonstrated that the sonographic parameters HC, BPD, AC, FL, and TCD had a strong positive monotonic relationship with gestational age (Figure 1A). The relationships between gestational age and FL/AC, TCD/AC, and HC/AC ratios were not monotonic and are visualized in Figure 1B. FL/AC remained fairly constant between 20 and 40 weeks of gestation, whereas HC/AC correlated negatively with gestational age. TCD/AC showed an undulating curve with a minimum at 20 weeks of gestation. The associated regression results are shown in Appendix A.

The binary logistic regression demonstrated for each individual sonographic parameter that an increase was associated with a decreased risk of SGA and FGR (see Figure 2A and Figure 3A). An increased FL/AC, HC/AC, or TCD/AC ratio was associated with an increased risk of SGA and FGR, respectively (see Figure 2B and Figure 3B). The results of logistic regression analysis for the gestational-age-dependent association between sonographic parameters and SGA and FGR are summarized in the supplements (see Appendix A). The exact results of gestational-age-dependent cutoff values for SGA and FGR, as shown in Figure 2A,B and Figure 3A,B, are presented in Appendix A. Appendix A summarize the accuracy of the cutoff values for predicting SGA and FGR fetuses, respectively.

The results for all parameters in the 39 + 0 weeks category were not coassessed because a too-small patient collective of 15 subjects proved to be unrepresentative. Comparing the tables and figures for SGA and FGR, it can be seen that the single parameters (HC, BPD, AC, FL, TCD) performed better than the quotients (FL/AC, TCD/AC, HC/AC,) in predicting SGA and FGR. However, AC most influenced SGA and FGR risk. At 34 + 0–38 + 6 weeks of gestation, the AC had a sensitivity, specificity, positive predictive value (PPV), negative predictive value (NPV), and Youden index (J) of 90%, 91%, 74%, 97%, and 81%, respectively for SGA fetuses and 98%, 98%, 82%, 100%, and 96%, respectively for FGR fetuses. For the sonographic parameters, the Youden index ranged from 0.11–0.81 in identifying SGAs and from 0.07–0.96 in identifying FGRs depending on the week of gestation. The best Youden index among the quotients was the HC/AC ratio, with 0.27 between 24 + 0–28 + 6 weeks of gestation for SGAs, and the TCD/AC ratio, with 0.31 between 24 + 0–28 + 6 weeks of gestation for FGRs. In the identification of SGAs, it can be observed that the Youden index of FL/AC reached only a value of 0.18 until 33 + 6 weeks of gestation, but between 34 + 0–38 + 6 had the highest value with 0.28 compared to the other quotients; the quotient of TCD/AC dropped to 0.09. FL/AC performed the worst in identifying FGRs compared to the other two quotients.

SGA and FGR were predicted with higher accuracy if a multivariable model including maternal age and weight, gestational age, BPD, HC, AC, FL, and TCD was used (Figure 4 and Figure 5).

In the multivariable model’s receiver operator curve (ROC) analysis, the areas under the curve were 0.950 and 0.965. In the multivariate model, an increased HC was associated with an increased risk of SGA (OR: 1.01; 95% CI 1.04–1.09; *p* < 0.001).

However, an increased value for all other parameters was associated with a lower risk of SGA (all OR < 1; all *p* < 0.001). Similar results were found for the prediction of FGR (Figure 5).

## 4. Discussion

The purpose of the present study was to compare the clinical utility of sonographic parameters and formulate cutoff values for predicting SGA and FGR. We could confirm with a large number of measurements that individual sonographic parameters predict SGA and FGR fetuses better than ratios. AC dominated as the single most informative sonographic parameter.

One of the most important goals of prenatal care is to assess fetal growth. FGR remains a complex obstetrical problem associated with an increased risk of adverse perinatal outcomes, such as premature birth, fetal hypoxia, neonatal acidosis, or intrauterine death. However, there is no universal consensus regarding the diagnostic criteria for FGR. As race, fetal sex, and geographic location influence average fetal growth, it is difficult to reach a universal consensus on a clinically useful definition for SGA and FGR [2].

In February 2018, the ACOG published a review that provides an overview of the different-country-specific guidelines for SGA and FGR [9]. The German guideline program defines SGA as an estimated fetal weight or birth weight < 10th percentile. FGR is defined as an (1) estimated fetal weight < 10th percentile and/or decrease of growth percentile during the course of pregnancy and (2) pathological Doppler ultrasound of the umbilical artery or uterine arteries or oligohydramnios [8]. SGA simply describes a weight at the lower end of the normal distribution, which is not necessarily associated with pathological growth. However, the lower the percentile, the higher the risk of morbidity and mortality, especially <3rd percentile [8]. In a large retrospective cohort study of more than three million singleton births, Pilliod et al. showed that the risk of stillbirth increased approximately three-fold for birth weight < 3rd percentile compared to the 3rd–5th percentile group and four- to seven-fold risk compared to the 5th–10th percentile group [28]. This prompted us to define FGR as estimated fetal weight and birth weight below the 3rd percentile. By only including cases with confirmed weight below the 3rd percentile, we excluded fetuses falsely classified as FGR. Conversely, if we had defined the infants by their birth weights only, as Beune et al. did, 450 neonates would have fit FGR criteria at birth, but would not have received growth monitoring due to their normal weight at the time of examination [29].

*Sonographic parameters:* An important prerequisite for diagnosing growth retardation is accurate pregnancy dating. Determining gestational age based on the last menstrual period may lead to incorrect assumptions. We decided to use ultrasound-based gestational age, which was defined by an early ultrasound measurement. In 1987, Campbell et al. published normative measurements of the cerebellum during pregnancy, making it possible to estimate the gestational age (positive correlation between TCD and menstrual age; R = 0.96) and enable the evaluation of abnormal fetal growth [30]. Due to the brain-sparing effect, the cerebellum is relatively resistant to hypoxic conditions; the blood supply to the brain is maintained at the expense of the systemic supply [10]. Thus, Reece et al. described TCD as a parameter minimally affected by growth restriction [11]. While this parameter is ideally suited for determining the gestational age, our analyses showed that the TCD is negligible for predicting both SGA and FGR fetuses. HC is another parameter only slightly affected by growth changes or external influences that could distort the fetal head. Our analysis showed that although HC performs slightly better in predicting SGA and FGR fetuses than BPD, it should not be used as the sole diagnostic tool. The AC reflects, amongst other organs, the size of the liver, which is affected early in the process of growth retardation due to glycogen depletion and correlates with the degree of fetal malnutrition [12]. As described by Baschat et al., an AC < 10th percentile has the highest sensitivity for diagnosing FGR [31]. We were able to confirm this in our study for both the SGA and FGR groups and found that sensitivity and specificity increased with gestational age. The FL is most commonly used to assess the fetus’s longitudinal growth. There are several proposed mechanisms (altered blood flow, altered secretion of bone-related growth factors) for the development of a short FL in the context of placental dysfunction [32]. Weisz et al. described a relationship between the isolated short femur and growth restriction. Compared to the other sonographic parameters, identification of SGA or FGR fetuses was better with FL than with HC or BPD, but worse than with AC. This may be due to the definition of an isolated short femur. The author defined a short femur as below the 5th percentile, whilst in our study, we defined an isolated short femur as <10th percentile for the SGA fetuses and as <3rd percentile for the FGR fetuses.

In addition to individual sonographic parameters, we calculated quotients based on a ratio between AC and a biometric parameter less sensitive to placental insufficiency. The most commonly used ratios are the FL/AC ratio, TCD/AC ratio and the HC/AC ratio.

*FL/AC ratio:* Hadlock et al. found that the FL/AC ratio was relatively constant throughout gestation in normal pregnancies, as was the case in our study. While a high FL/AC ratio (>90th percentile) has been associated with a birth weight < 10th percentile, the association is weak [12]. Our evaluation showed an overlap of the FL/AC ratio in normal and growth restricted fetuses without establishing a reasonable cutoff value to identify SGA or FGR. While for FGR, the sensitivity was high, specificity and thus Youden’s index were low. One reason could be the small number of SGA and FGR cases compared to AGA (Table 1).

*TCD/AC ratio:* In theory, the TCD/AC-ratio is a perfect parameter for the detection of FGR, as chronic fetal hypoxia affects the TCD less than other biometric parameters and the AC is a more sensitive parameter for fetal growth than BPD or FL. This is particularly the case in fetuses of uncertain gestational age, as the TCD/AC ratio remained relatively constant throughout gestation [13,14,17,33,34]. In a prospective study of 825 low-risk pregnancies and 158 pregnancies at risk of growth retardation, Meyer et al. demonstrated that the TCD/AC ratio has a high sensitivity (83.9%) for SGA and noted that the TCD/AC ratio is an accurate, gestational-age-independent method of identifying SGA infants [26]. In our study, the cutoff value of TCD/AC-ratio was found to have a relatively low total sensitivity of 23% and a total specificity of 94% for identification of SGA—30% and 97%, respectively, for identification of FGR. Dilmen et al. calculated a sensitivity of almost 100% for the TCD/AC ratio cutoff value; however, only 10 fetuses with FGR were examined, which calls the significance into question [34].

Some authors have pointed out that as a weakness of using the TCD/AC ratio, the interval between the examination and the time of birth should not be too great in order to detect FGR accurately [15,16,26,33,35]. To eliminate this negative influencing factor, we only included fetuses that had both an estimated weight and birth weight < 10th and <3rd percentile, respectively, representing 100% of the growth restricted fetuses. In our analysis, the specificity and NPV (0.94 and 0.90 for SGA; 0.97 and 0.98 for FGR) were very high because the SGA rate—and especially the FGR rate—in our dataset was low compared to the AGA fetuses. However, despite their high specificity, the factors are not well-suited. These results are consistent with the regression results, where the AUC was also low. Comparing the diagnostic value of previous and current cutoff data, the significance is inconclusive and difficult to compare with other studies due to smaller case numbers and different cutoff values [15,17,26,33,34,35]. The diverse populations studied could explain the variance between TCD/AC ratios, resulting in differing cutoff values.

*HC/AC ratio:* Campbell et al., 1977 first investigated the relationship between HC and AC to determine whether the HC/AC ratio was clinically important in distinguishing between symmetrical and asymmetrical growth retardation [27]. The use of the HC/AC ratio in classifying symmetric and asymmetric growth retardation is controversial. Dashe et al. reported an association between an increased HC/AC ratio (>95th percentile), indicative of an asymmetrical pattern, and adverse pregnancy outcomes [36]. In contrast, Colley et al. postulated that the ratio is of limited clinical use and confirmed this in a study of 999 infants, in which the HC/AC ratio correlated poorly with the Ponderal index (PI) [18]. In newborns, the PI enables the distinction between symmetric and asymmetric FGR and measures the severity of asymmetry in growth-restricted newborns. We did not distinguish between asymmetric and symmetric FGR in this study. The cutoff value of the HC/AC quotient seems to discriminate well between growth restricted and non-growth-restricted fetuses with high total specificity in SGA fetuses of 91%. However, the fact that the quotient is not well-suited is shown by the low sensitivity, low PPV, and low Youden index of 31%, 31%, and 22%, respectively. Similarly unfavorable results for the identification of FGR were achieved.

*Multivariable Model:* Besides ultrasound parameters, other factors such as maternal diseases, maternal height, smoking, or drug use can influence fetal growth. In particular, race and maternal age have been shown to be risk factors for growth retardation [2]. In addition to studying the individual sonographic parameters and ratios, we attempted to increase the detection rate by using a multiregression analysis and obtained an excellent result with this analysis. Not only sonographic parameters of the fetus but also maternal characteristics were included. Obviously, the more factors one includes, the better results are obtained, but as both maternal age and weight are extreme risk factors for FGR, we restricted ourselves to these two. With an AUC of 0.950 and 0.965, respectively, the hit rate in this analysis was just below 100% for both SGA and FGR identification.

One limitation of this study is the unequal distribution of gestational ages with more than half of the examinations performed between 19 + 0 and 23 + 6 weeks of gestation. Different sample sizes in the respective study periods naturally limit the transferability of these results. However, the larger number of subjects of early gestational age improves the significance of the results in the early weeks of pregnancy and reflects clinical practice regarding the second trimester screening. Early detection of growth retardation is essential, as appropriate monitoring can be established earlier, lessening morbidity and mortality.

A further limitation of this study is that growth restriction is not limited to fetuses with an EFW < 10th percentile. However, these fetuses present additional abnormalities, such as pathological Doppler ultrasound, oligohydramnios, or lack of interval growth, which were not assessed in this study.

When comparing the results of this study with other studies, differences may be due to sample size, ethnicity, gestational period, and selection bias.

## 5. Conclusions

The individual sonographic parameters are more suitable for identifying SGA and FGR than their ratios. Among the individual parameters, AC proved to be the most suitable, especially in advanced weeks of gestation (34 + 0–38 + 0). It is impossible to define a clear ratio threshold, as overlaps are too large. However, a near-perfect detection with high accuracy is obtained if a multivariable model is used. FGR should be distinguished from constitutionally small SGA fetuses, as in former morbidity and mortality increases. Our research illustrates the difficulty of identifying growth-restricted fetuses based on biometric parameters alone and indicates that the diagnosis should be based on a combination of sonographic parameters and Doppler studies.

Obstetricians who are not familiar with detailed Doppler ultrasound may have difficulty diagnosing FGR and could use the multivariable model to calculate the probability of SGA or FGR. The more factors included in a multivariable analysis, the better. The ones we have included are the most practical, are easy to determine, and produce representative results. Although there is no prenatal treatment, fetal monitoring can be initiated after diagnosis and the timing of delivery can be optimized to avoid stillbirths.

## Figures and Tables

**Figure 1 jpm-12-01125-f001:**
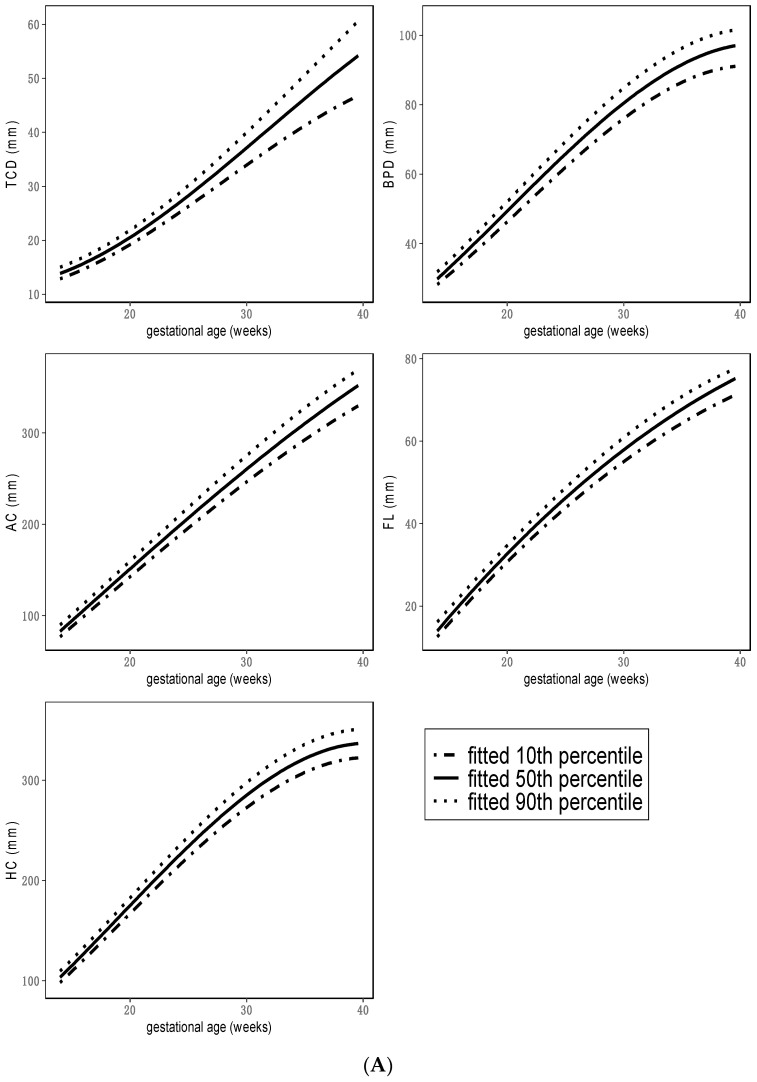
(**A**) Reference curves for the ultrasound parameters within this study population. Results of all quantile regression models are presented in Appendix A. AC = abdominal circumference, BPD = biparietal diameter, FL = femur length, HC = head circumference, TCD = transcerebellar diameter. (**B**) Reference curves for the ultrasound ratios within this study population. Results of all quantile regression models are presented in Appendix A. FL/AC = femur length/abdominal circumference, HC/AC = head circumference/abdominal circumference, TCD/AC = transcerebellar circumference/abdominal circumference.

**Figure 2 jpm-12-01125-f002:**
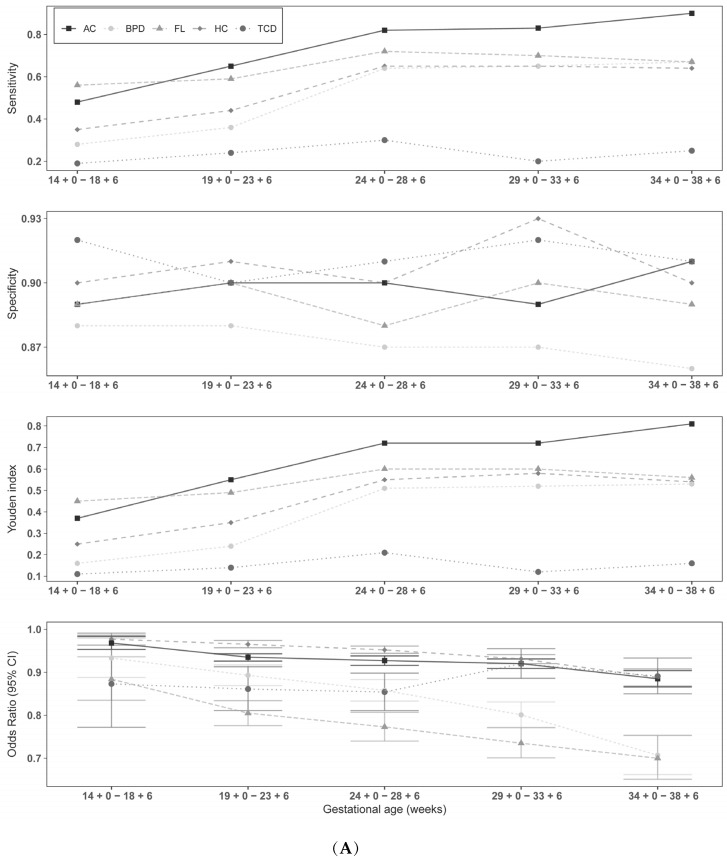
(**A**) Results of ultrasound variable cutoff values for detection of SGA. Results of gestational-age-dependent cutoff values are presented in Appendix A. AC = abdominal circumference, BPD = biparietal diameter, FL = femur length, HC = head circumference, TCD = transcerebellar diameter. (**B**) Results of the ultrasound ratio for SGA. Results of gestational-age-dependent cutoff values are presented in Appendix A. FL/AC = femur length/abdominal circumference, HC/AC = head circumference/abdominal circumference, TCD/AC = transcerebellar circumference/abdominal circumference.

**Figure 3 jpm-12-01125-f003:**
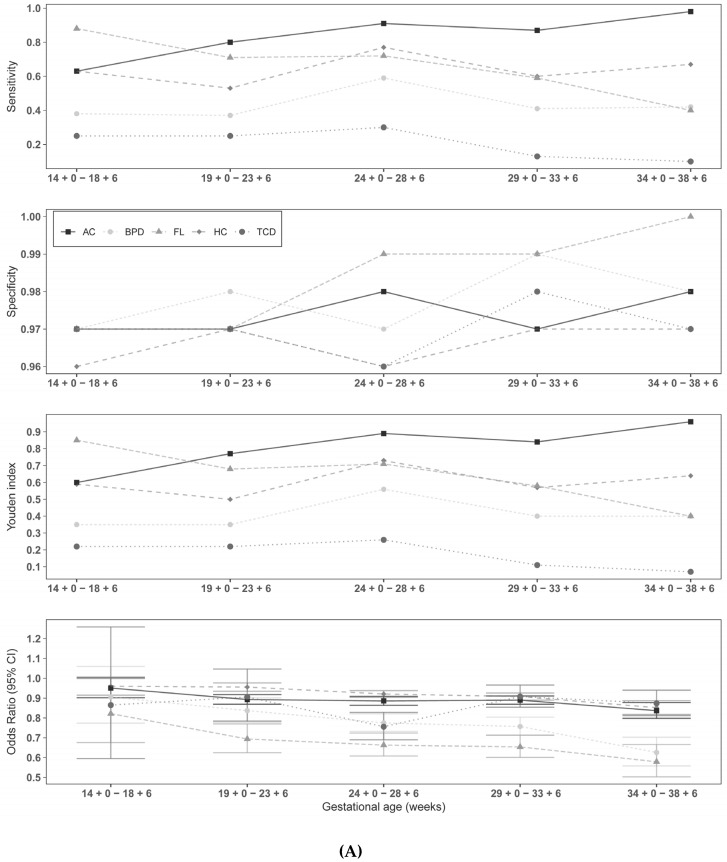
(**A**) Results of ultrasound variable cutoff values for detection of FGR. Results of gestational-age-dependent cutoff values are presented in Appendix A. AC = abdominal circumference, BPD = biparietal diameter, FL = femur length, HC = head circumference, TCD = transcerebellar diameter. (**B**) Results of ultrasound ratio cutoff values for detection of FGR. Results of gestational-age-dependent cutoff values are presented in Appendix A. FL/AC = femur length/abdominal circumference, HC/AC = head circumference/abdominal circumference, TCD/AC = transcerebellar circumference/abdominal circumference.

**Figure 4 jpm-12-01125-f004:**
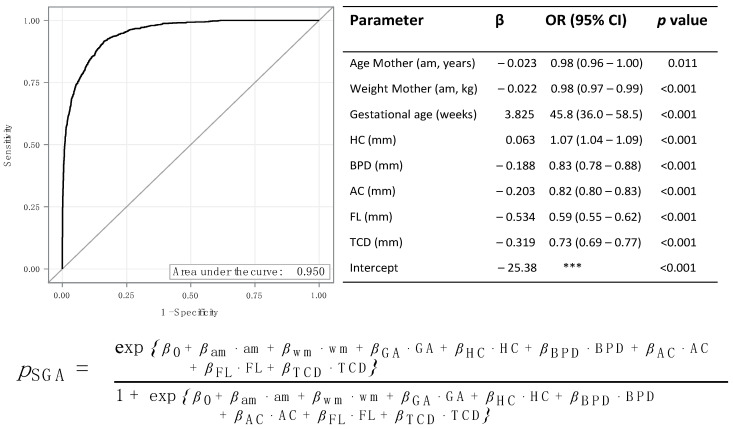
Multivariable regression: Prediction of SGA. AC = abdominal circumference, BPD = biparietal diameter, *ß* = regression coefficient, CI = confidence interval, FL = femur length, HC = head circumference, SGA = small for gestational age, OR = odds ratio, TCD = transcerebellar diameter. *** CI not given.

**Figure 5 jpm-12-01125-f005:**
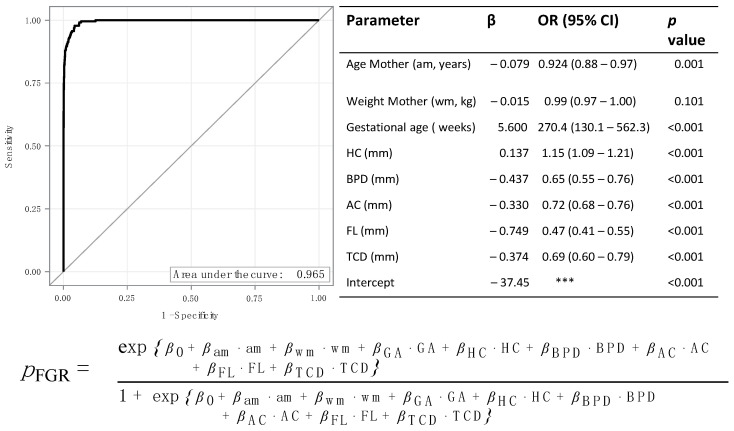
Multivariable regression: Prediction of FGR. AC = abdominal circumference, BPD = biparietal diameter, *ß* = regression coefficient, CI = confidence interval, FGR = fetal growth restriction, FL = femur length, HC = head circumference, OR = odds ratio, TCD = transcerebellar diameter. *** CI not given.

**Table 1 jpm-12-01125-t001:** Characteristics of the study population.

	Median (Range)	N (%)			
Examinations		9292 (100)			
*Maternal characteristics*					
Age, years	33 (14–50)				
BMI	25 (15–65)				
*Fetal characteristics*					
Gender					
Male		4765 (51)			
Female		4527 (49)			
Estimated fetal weight, g	426 (91–4091)				
Birth weight, g	3370 (172–4390)				
AGA		8231 (89)			
SGA		1061 (11)			
FGR		255 (24)			
Subdivision of the gestational age (weeks)		Total	AGA	SGA	FGR
14 + 0–18 + 6		1299 (14)	1219 (15)	80 (8)	8 (3)
19 + 0–23 + 6		4914 (53)	4517 (55)	397 (37)	59 (23)
24 + 0–28 + 6		1221 (13)	1012 (12)	209 (20)	64 (25)
29 + 0–33 + 6		1212 (13)	978 (12)	234 (22)	75 (29)
34 + 0–38 + 6		631 (7)	494 (6)	137 (13)	48 (19)
≥ 39 + 0		15 (0)	11 (0)	4 (0)	1 (0)
GA ultrasound, weeks	21 (14–40)				
GA delivery, weeks	39 (19–43)				

Acronyms: AGA = appropriate-for-gestational age, BMI = body mass index, FGR = fetal growth restriction, GA = gestational age, SGA = small-for-gestational age.

## Data Availability

The data analyzed were retrieved from the university hospital of muenster database.

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
