# Peer review of "Fetal Growth Restriction: Comparison of Biometric Parameters"

_jpm, 2022, doi:10.3390/jpm12071125_

Round 1

Reviewer 1 Report

Dear Authors,

I have reviewed and largely appreciated the manuscript titled Fetal growth restriction: Comparison of biometric parameters a well-focused and coherently constructed piece of research aimed at assessing tthe clinical utility of sonographic parameters and formulate cut-off values for predicting  small for gestational age (SGA) and fetal growth restriction (FGR). The article has been competently and thoroughly delineated to pursue its goal of comparing various biometric criteria for the purpose of identifying growth retardation and evaluate the clinical value of standard ultrasound ratios such as FL/AC, TCD/AC, HC/AC. Furthermore, the authors aimed to create a regression model for predicting SGA/FGR in order to heighten current prenatal detection rate. I feel the chief and most meaningful strengths of this article reside in its thoroughness and rigor. It relies on sound methodology in addition to straightforward and well-conceived tables and figures. The study has accounted for the biometric data of 9292 pregnancies between the 15th and 42nd weeks of gestation, and the statistical analysis hinges on descriptive data, quantile regression estimating the 10th and 90th percentiles and multivariable analysis.

The study's weakness that needs addressing is that it does not succeed in providing enough background and contextualization as to the relevance of an accurate assessment of FGR, which detracts from overall interest and relevance to a broad scientific readership. Discussion and Conclusions need broader elaboration as to the clinical implications and maternal risk factors affecting FGR rates.

The article is very well written, but I would still recommend further proofreading from a native speaker of English in order to streamline the prose overall, which at times comes across as a slightly convoluted and clumsy.

Sincerely,

Author Response

Dear Reviewer,

thank you for reviewing the paper so carefully and appreciating its thoroughness and rigor.

Thank you also for your constructive criticism regarding the contextualization of the relevance of an accurate assessment of FGR. We have revised the introduction, discussion and conclusion in this regard accordingly. In both the discussion and conclusion, we have gone into greater detail regarding the maternal risk factors that influence the FGR rate. Furthermore,  the paper was reviewed by a native speaker in order to streamline the prose. We hope you now find it more satisfactory.

At the suggestion of another reviewer, we decided not to include Tables 2 and 3 in the results but in the supplementary material to improve comprehensibility.  This way, if other authors later want to compare their data with ours, they can consult the supplements.

Sincerely,

Carolin Marchand

Reviewer 2 Report

This study, aimed to compare the sonographic parameters for identification of SGA/FGR. 

Major comment: 

Initially, I perceived the authors have aimed to explore the sonographic parameters and if there could be a strategy to predict growth restriction (pathologic) vs SGA( non-pathology). And this seemed novel, interesting and of importance. I am still not sure if this was the initial aim of the authors or not, but what this manuscript currently presents is the standard sonographic parameters and their predictive value for SGA and FGR without any specific demarkation between the two. Furthermore, I am not sure if FGR is defined very accurately (as far as I understand it't not the percentile and is "restriction" in growth/ falling off the expected curve. 

Other comments: 

TCD is not spelled out

the fetal ultrasounds are done at variable gestational age with limited sample size at specific gestational age for the population of interest. Needs to be acknowledged. 

The median and range of weight goes up to 4.3kg. If LGA is defined as >90th percentile, this is falling in LGa category and authors have said this population were excluded. 

In the result sections, there are so many graphs and tables. Still not addressing the interesting/important questions. Suggest to remove Figure 1A. For figure 2 and table 3, one should be adequate and authors are encouraged to try addressing the question of SGA vs FGR prediction using the parameters. 

Author Response

Dear Reviewer,

thank you for taking the trouble to review the paper and for your detailed and constructive criticism. We have addressed your comments in our working group below.

  1. Aim of the study

Our aim was to review sonographic parameters and ratios in relation to fetal growth retardation and assess whether it is possible to define cut-off values by which to identify growth retarded fetuses. However, it was not our aim to distinguish between SGA and FGR.

The German guideline for intrauterine growth restriction states that up to 70% of

fetuses below the 10th percentile are constitutionally small fetuses and have a normal perinatal outcome, whereas FGR is associated with greater perinatal morbidity and mortality and can be identified by additional abnormalities (pathological Doppler sonography, oligohydramnios, lack of interval growth or an estimated weight < 3rd  percentile). To date, there is no standard international definition for FGR fetuses. This was most recently elaborated in 2018 by McCowan et al. [1]. Just as you described (and in accordance with the German guidelines), a deviation of the fetal weight from the percentile curves, denoting insufficient interval growth, can be used to define FGR. However, as a longitudinal assessment of fetal growth was not part of this study – we used only one measurement per fetus – we defined FGR as estimated and birth weight < 3rd percentile. The introduction has been revised for more clarity.

  1. TCD is not spelled out

Thank you for pointing this out, it has been corrected accordingly.

  1. Ultrasounds at variable gestational age with limited sample size

Thank you for pointing this out, it is an important limitation. We noted the different sample sizes in the examination periods in lines 122-124. In daily practice, screening of second trimester fetuses is performed between 20 and 24 weeks. Accordingly, 53% of patients were also screened between 19+0 and 23+6 weeks of gestation in our center. Different sample sizes in a respective study periods of course pose a significant limitation.

However, the distribution also reflects the clinical reality and the high number of subjects in the typical second trimester screening interval improves the validity of the results within this time period.

  1. The median and range of weight

We studied fetuses between 14+0 and 41+6 weeks of gestation. The fetus weighing 4390g was male and born at 41+2 weeks gestation. Fetuses were classified according to their birth weight into SGA/ AGA and LGA based on sex using the percentile values of birth weights according to Voigt et al [2]. According to Voigt et al, male neonates are at the 90th percentile from a birth weight of 4395g, and female neonates are at the 90th percentile from a weight of 4208g. Thus, this neonate is considered appropriate for gestational age. The LGA category was excluded, as you also wrote above.

  1. Graphs and tables in the result sections

Thank you for your constructive criticism regarding the graphs and tables. Figure 1A and 1B show the reference curves for the ultrasound parameters and ultrasound ratios, respectively. In our eyes, both graphs are relevant as the variables and ratios change during gestation.

If we delete Figure 1A, we would not be able to see the relationship between the weeks of gestation and the individual parameters.

Figure 2 and 3 show the results of ultrasound variable cut-off values for detection of SGA and FGR. Another reviewer noted that the figures and tables were well listed in the Results section. Nevertheless, we decided to include Table 2 and 3 in the supplemental material rather than in the results, in order to improve comprehensibility.  If other authors later wish to compare their data with ours, they can thus consult the supplements.

Overall, as recommended by you, we have gone into more detail about the results in the text and described them better. 

Sincerely,

Carolin Marchand

[1]      McCowan L, Figueras F, Anderson N. Evidence-based national guidelines for the management of suspected fetal growth restriction: comparison, consensus, and controversy. Am J Obstet Gynecol 2018; 218: S855–S868. doi:10.1016/j.ajog.2017.12.004

Round 2

Reviewer 2 Report

No further comments